# Synthesis of Integrated Flower Waste Biorefinery: Multi-Objective Optimisation with Economic and Environmental Consideration

Emily Hau Yan Chong [1], Viknesh Andiappan [1,2], Lik Yin Ng [1,3], Parimala Shivaprasad [4] and Denny K. S. Ng [1,5,*]

1. School of Engineering and Physical Sciences, Heriot-Watt University Malaysia, Putrajaya 62200, Malaysia
2. Research Centre of Sustainable Technologies, Faculty of Engineering, Computing and Science, Swinburne University of Technology Sarawak, Kuching 93350, Malaysia
3. Faculty of Engineering, Technology & Built Environment, UCSI University, Puncak Menara Gading, Taman Connaught, Cheras, Kuala Lumpur 56000, Malaysia
4. Faculty of Engineering, University of Nottingham, Nottingham NG 7 2RD, UK
5. Centre of Hydrogen Energy (CHE), Universiti Teknologi Malaysia (UTM), Johor Bahru 81310, Malaysia
* Correspondence: denny.ng@hw.ac.uk; Tel.: +60-3-88943784

**Abstract:** The improper disposal of flower waste from cultural activities is one of the main challenges in certain countries such as India. If the flower waste is not managed properly, it causes a number of environmental issues. Therefore, various technologies have been developed to transform flower waste into value-added products. To integrate multiple technologies holistically to maximise the energy and material recovery, an integrated flower-waste biorefinery is required. Since there are a wide range of technologies available that can convert the waste into multiple products, there is a need to develop a systematic approach to evaluate all the technologies. This research proposes a systematic approach to synthesise an integrated flower-waste biorefinery based on different optimisation objectives, e.g., maximum economic performance and minimum environmental impact. Due to the conflicting nature between the two objectives, a fuzzy optimisation approach has been adapted to synthesise a sustainable integrated flower-waste biorefinery that satisfies both objectives at once. The efficacy of the proposed approach is demonstrated through a case study in India based on the optimised results with fuzzy optimisation—a synthesised flower-waste integrated biorefinery with economy performance of $400,932 and carbon emission of 46,209 kg $CO_2$/h.

**Keywords:** flower waste; integrated biorefinery; waste valorisation; value-added product; mathematical optimisation

## 1. Introduction

Disposal of large amounts of flower waste is one of the major concerns that leads to environmental issues. Cultural and heritage activities, especially in India, have produced the highest flower waste worldwide [1]. India has a population of religious devotees who commonly practice religious customs using flowers, as they are tokens of devotion in South Asian cultures [2]. Thus, large amounts of flowers are left as offerings and accumulate at religious sites like temples. According to Statista 2021, India's flower production reached about 3 million metric tonnes in the financial year 2021 [1]. Meanwhile, Varanasi is the holiest city in India, which generates around 3.5 to 4 tonnes of flower waste from temples daily. Other than religious activities, flower waste is also generated during different ceremonies, functions and festivals. Most of the flowers utilised in these activities are left unused, resulting in a common source of waste. Viewing the issues above, it is important to have an effective waste disposal policy for the recovery of such waste, with an aim towards reducing the environmental impacts. In current practice, most wilted and unused flowers in India are discarded into landfills or water bodies [3]. This causes water pollution as toxic

pesticides and insecticides remaining in the flower waste would seep into the water bodies. In addition, organic matter in flower waste will cause negative impacts on water quality by depleting oxygen levels in water, threatening the marine ecosystem.

To address the above-mentioned issues, flower waste can be recovered and converted into value-added products. Recently, various technologies have been developed to recover organic waste materials and convert them into value-added products. To maximise the material and energy recovery potential, an integrated flower-waste biorefinery that integrates multiple technologies will be introduced. Detailed discussions on the potential of flower-waste valorisation and synthesis of a sustainable integrated flower-waste biorefinery will be presented in the following sub-sections.

### 1.1. Valorisation of Flower Waste

Waste valorisation is an application of industrial ecology that allows closed-loop manufacturing facilities to gain the benefits of the cyclic pattern of the used material present in the consumption sectors [4]. The waste generated by using flowers for decorative or religious purposes can be converted into value-added products for different applications: agricultural, bioenergy, food, beverage, pharmaceutical, etc. For example, phenolic compounds present in flowers could provide a high antioxidant capacity compared to fruits and vegetables. It provides positive effects on oxidative stress-related diseases such as cardiovascular diseases, neurodegenerative diseases, cancer, diabetes, obesity, and liver diseases [5]. Hence, to conserve the medical constituents in flowers, a proper drying method is required. Drying can improve the edible nature of flowers and prolong their shelf life. The dry-flower industry has the great potential to provide employment opportunities to thousands of people, especially to housewives and rural women in India, as unlimited aesthetic value and decorative products can be created by using the dry-flower technology [6]. In addition to being utilised for food and pharmaceutical applications [7,8], flower waste can be processed through different technologies for a variety of purposes. Slavov et al. [9] and Dutta et al. [10] reviewed valorisation opportunities of flower waste to essential oil, recovery of valuable biologically active substances, biofuels production, activated carbon and their application to food industry or medicine. Research work on investigating various strategies for recovery of flower waste have been done over the past years. Table 1 summarises the possible technologies for the utilisation of flower waste.

**Table 1.** Possible technologies for utilisation of flower waste.

| Application | Technology/Biorefinery | Description | Final Product |
|---|---|---|---|
| Food | Drying [7] | Removal of water from flower and retain the medical constituents | Edible flower and flower for garnishing purpose |
| Pharmaceutical | Extraction [8] | Secure biological active compound in flower | Flower for medical purpose |
| Agricultural application | Composting [9] | Utilised for soil replenishment after composting | Bio-fertilizers |
| Environmental remediation | Biosorption [9] | Heavy metals from wastewater are accumulated | Removal of dyes, wastewater treatment |
| Bioenergy application | Gasification and fermentation [9] | Turn waste into source of energy | Biogas and bioethanol |

### 1.2. Synthesis of Sustainable Integrated Biorefinery

According to Kamm (1997), a biorefinery is defined as a complex system of sustainable, environmental and resources-friendly technologies for fully utilising and exploiting biological raw materials [11]. All technologies stated in Table 1 may be applied to convert biological raw materials into value-added products. Research based on different types of biological raw materials has been conducted in the past. To maximise the energy and

material recovery, an integrated biorefinery was introduced [12]. An integrated biorefinery consists of multiple technologies including feedstock handling, pretreatment processes, and different biomass conversion (e.g., thermochemical, biological, physical, etc.) and upgrading processes to produce value-added products and energy [13].

Viewing the importance of sustainable production and consumption in the industry, systematic synthesis and optimisation of integration biorefinery have been developed since the last decade. A number of systematic tools have been developed, ranging from hierarchical approaches [14,15], algorithmic approaches [16,17], mathematical optimisation approaches and hybrid approaches. For example, Ng et al. [14] proposed a hierarchical approach known as the forward-reverse synthesis tree to synthesise and analyse integrated biorefineries. Most recently, Tey et al. [15] presented an extended hierarchical decomposition approach, which was developed for chemical process synthesis for the synthesis of biorefinery processes. Based on the benefit of algorithmic approaches to execute a sequential set of actions based on automated reasoning, calculation and data processing, P-graph was adapted to synthesise an integrated biorefinery [16] and integrated palm-based biorefinery based on the circular economy concept [17]. To address a complex synthesis problem with multi-criteria requirements, mathematical optimisation approaches have been developed. For example, a superstructure-based shortcut approach was developed by Bao et al. for the conceptual design of integrated biorefineries [18]. A superstructure optimisation model was then adapted in the synthesis of integrated biorefinery for different feedstock, such as a palm-based biorefinery [19], microalgal biorefinery [20], seaweed-based biorefinery [21], etc. A detailed review of the synthesis tools can be found in the recent review papers by Chemmangattuvalappil et al. [22].

Environmental Assessment on Integrated Flower-Waste Biorefinery

To synthesise a sustainable integrated biorefinery, an environmental assessment is one of the important factors. As shown by Romero-García et al. [23], various methods can be adopted to evaluate the environmental impact of an integrated biorefinery. These include life cycle assessment (LCA), potential environmental impact (PEI), greenhouse gases (GHG) emission, footprints (carbon, water and sustainability), etc. LCA is a systematic, scientific method for calculating and evaluating long-term environmental impact [24]. LCA has been applied in various integrated biorefineries such as cyclamen plants in greenhouse cultivation [25] and Ethiopian rose cultivation [26]. Russo et al. [25] applied LCA towards roses and cyclamens in greenhouse cultivation. According to Saraiva [27], there are a number of parameters that need to be considered when doing life-cycle assessment, as shown in Table 2.

**Table 2.** Parameters considered in measuring environmental impacts.

| Parameters | | Consideration |
|---|---|---|
| Direct environment impacts | Input-related emission | Greenhouse gases emissions from crop production |
| | Transportation of feedstock | Furl use for feedstock collection and transportation of residues and by-product |
| | Direct land-use change | Soil quality changes made by above-ground biomass in terms of carbon sequestration |
| Indirect environmental impacts | Indirect land-use change | Substitution of one feed into another feed will affect greenhouse gases emission (this section is not elaborated further in journal review) |

Alig et al. [28] applied LCA to determine the environmental impacts of production of different types of cut roses from Holland, Kenya and Ecuador. In the previous work [28], the emission of gases from the packaging and transportation of cut roses are considered when performing LCA.

Limited research has been done on the synthesis of a flower-waste integrated biorefinery. Thus, this work aims to synthesise an integrated flower-waste biorefinery. To design a sustainable integrated flower-waste biorefinery, other than an environmental assessment, the economic performance of the synthesised biorefinery should also be considered. Therefore, such a design problem is required to be solved via multi-objective optimisation approaches. Note that a number of works have been developed to synthesise an integrated biorefinery with multi-objective functions, such as trade-off between environmental and economic performances [29–32], under uncertainty [33], chemical product design [34], inherent safety and health [35], etc.

According to Tay and Ng [29], fuzzy optimisation identifies the ideal alternatives in decision-making problems by solving an objective function on a set of alternatives given by fixed constraints; the preferable alternatives have the more desired maximum or minimum objective function values. Thus, a fuzzy optimisation is adapted in this work.

## 2. Problem Statement

Given a set of flower waste based on various compositions (i.e., petal, leaf, stem), $i \in I$ $\left(F_i^{\text{Biomass}}\right)$ that may be transformed into intermediates $k \in K$ and then further converted into products $k' \in K'$ via technologies $j$ and $j'$ based on fixed conversion factors, these conversions factors for intermediates $k$ and final product $k'$ are denoted as $X_{ijk}$ and $X_{kj'k'}$. Based on the conversion factors, the flowrates $F$ of each intermediate $k$ and product $k'$ can then be determined. A generic superstructure of the model is illustrated in Figure 1.

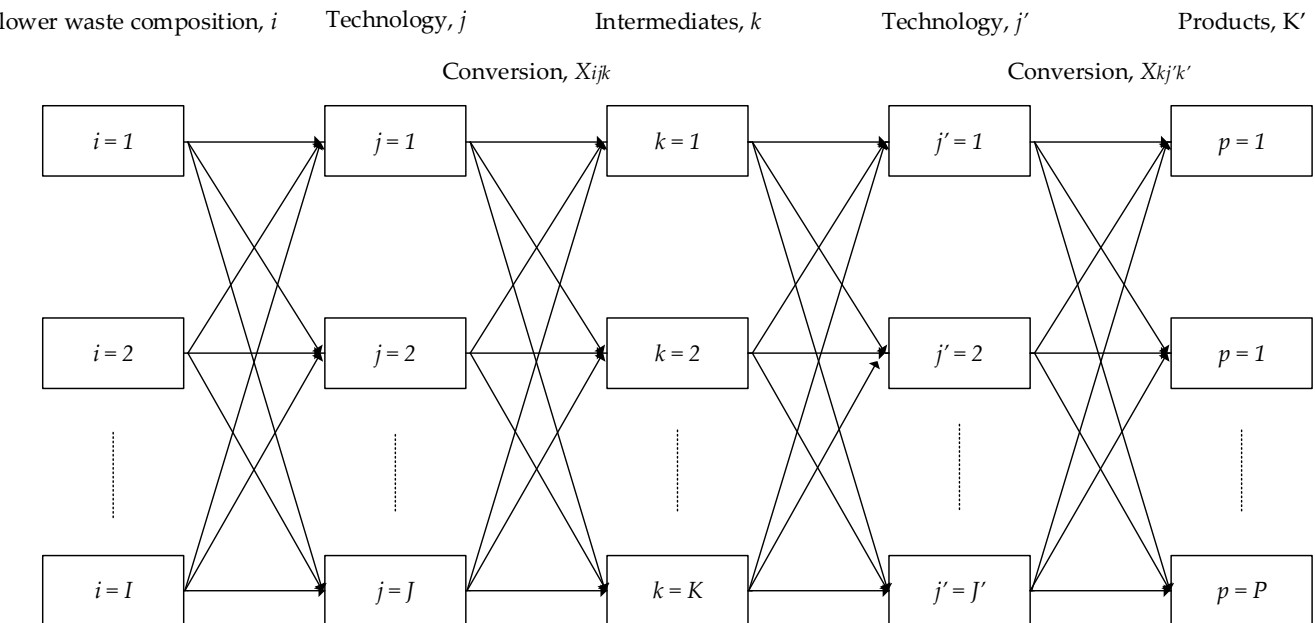

**Figure 1.** Generic superstructure for flower-waste biorefinery.

The objective of this work is to produce an integrated flower-waste biorefinery with economic and environmental considerations. Three scenarios are analysed in the case study: (i) maximum economic performance, (ii) minimum environmental impact and (iii) multi-objective optimisation with maximum economic performance and minimum environmental impact.

## 3. Mathematical Optimisation Model

Before formulating the mathematical optimisation model, data collection (e.g., conversion performance, emission factor, material and energy requirement, price of product and cost of materials, etc.) for all possible technologies and processes that valorise flower waste into value-added products were collected. Then, a superstructure model that includes all

possible process alternatives was generated and a mathematical optimisation model was developed based on different optimisation objectives.

*Model Formulation*

Referring to Figure 1, the total amount of flower waste $i$ is first sent to various biomass conversion technologies $j$ to produce intermediates $k$. The intermediate then undergoes upgrading process $j'$ to be further converted into products $p$ or $K'$. The mass flow distribution within the integrated biorefinery is represented by Equation (1).

$$F_i^{\text{Biomass}} = \sum_j f_{ij} \ \forall \ i \tag{1}$$

where $f_{ij}$ is the mass flowrate of the flower waste to technologies $j$.

$$F_k = \sum_j \sum_i f_{ij} X_{ijk} \ \forall \ k \tag{2}$$

where $X_{ijk}$ is the conversion factor of flower waste $i$ to intermediate $k$.

For intermediate $k$, which requires further processing in technologies $j'$, the mass flowrate of final product $k'$ is given as below:

$$F_k = \sum_{j'} f_{kj'} \ \forall \ k \tag{3}$$

where $f_{kj'}$ is the mass flowrate of flower waste intermediate to technologies $j'$.

$$F_{k'} = \sum_{j'} \sum_k f_{kj'} X_{kj'k'} \ \forall \ k' \tag{4}$$

where $X_{kj'k'}$ $X_{kj'k}$ is conversion factor of intermediate $k$ to final product $k'$.

Note that the above mathematical model can be modified to suit different technologies $j$ and $j'$. For example, in the drying process, water within the biomass conversion is performed to determine the water lost during the process and total mass changed. For the technologies that convert the biological material into bioenergy, the conversion factor X can be replaced by an energy conversion factor.

Other than tracking the final products, waste generated throughout the biorefinery is also determined. In this work, it is assumed that all the waste that cannot be converted into value-added products will be sent to a landfill. The equation below describes the mass flow of the collected waste to a landfill, from intermediates $k$ and final product $k'$ in technologies $j$ and $j'$.

$$F^{\text{Landfill}} = \sum_j \sum_i f_{ij}\left(1 - X_{ijk}\right) + \sum_{j'} \sum_k f_{kj'}\left(1 - X_{kj'k'}\right) \tag{5}$$

The economic performance of the integrated biorefinery (*EP*), the revenue (*REV*) and landfill cost (*COST*$^{\text{Landfill}}$) are determined as below:

$$REV = \sum_k F_k \times P_k + \sum_{k'} F_{k'} \times P_{k'} \tag{6}$$

$$COST^{\text{Landfill}} = F^{\text{Landfill}} \times \text{TR} \tag{7}$$

$$EP = REV - COST^{\text{Landfill}} \tag{8}$$

where $P_k$ and $P_{k'}$, TR are the selling price of intermediate $k$, final product $k'$ and disposure cost, respectively.

Environmental impact is one of the crucial factors when developing a process. In this work, the parameters that affect the total environmental impact $IMP^{\text{Total}}$ are the energy required for different technologies and carbon emissions based on one unit of electricity usage. The overall environmental impact in this work is determined via Equation (9):

$$IMP^{\text{Total}} = \sum_i \sum_j T_j \times \text{CE}_{ijk} + \sum_k \sum_{j'} T_{j'} \times \text{CE}_{kj'k'} \tag{9}$$

where $T_j$ and $T_{j'}$ is the energy required for technologies in the biomass conversion process and upgrading process, while $CE_{ijk}$ and $CE_{kj'k'}$ are the carbon emissions when one unit of electricity is utilised in technologies $j$ and $j'$.

## 4. Case Study

A case study from India has been used to illustrate the proposed systematic optimisation approach. As mentioned previously, valorisation of flower waste in an integrated biorefinery involves various types of technologies to transform flower waste into valuable product. Based on a detailed literature review, all available technology pathways such as drying, extraction and production to generate bioenergy are summarised in Table 3. Note that the technologies can be widely divided into three categories, which are drying, extraction and generation of bioenergy.

**Table 3.** Summary of flower-waste biorefineries.

| Technology/Biorefinery | | Description |
|---|---|---|
| Drying | Press Drying [36] | Apply pressure while drying |
| | Sun Drying [8] | Natural drying with the aid of sun |
| | Shade Drying [37] | Natural drying without direct sunlight |
| | Embedded Drying [7] | Silica gel as desiccants for dying |
| | Freeze Drying [7] | Sublimation process in low temperature |
| | Glycerine Drying [7] | Glycerol solution as medium to absorb water |
| Extraction | Hydrodistillation [38] | Extraction process involving direct heat boiling |
| | Hydrolysis [38] | Chemical reaction to form acid and alcohol |
| | Supercritical Carbon Dioxide [38] | Carbon dioxide extraction in supercritical state |
| | Phytosol Extraction [38] | Cold extraction without heat |
| | Simultaneous Distillation [38] | Extraction with gas chromatography with two flasks |
| | Membrane Extraction [38] | Extraction involves partitioning in a membrane |
| | Fermentation [39] | Chemical breakdown of flower waste |
| Bioenergy Generation and Transesterification Reaction | Composting [40] | Utilised for soil replenishment after composting |
| | Biosorbent [41] | Heavy metals from wastewater are accumulated |
| | Gasification [42] | Convert flower waste into gas in high temperature |
| | Transesterification [43] | Convert into geranyl acetate and citronellyl acetate |

### 4.1. Superstructure

Based on data collected as shown in Table 3, all possible technology pathways are collected and presented in a superstructure shown in Figure 2. For easier comprehension, the technologies are divided into different categories based on the function of the technologies, namely drying, extraction and bioenergy generation. Note that the superstructure can be divided into several parts: first pretreatment of flower waste, a biomass conversion process that converts the pretreated flower waste into intermediates, and the intermediates are then further converted into final products via upgrading processes.

In this work, the moisture content of flower waste is identified as the major limitation in the main process; therefore, drying processes are taken as pretreatment systems. Based on a literature review, the drying process allows for the reduction in moisture content in flower waste from 40 wt% to 10 wt% [36]. As a result, a two-stage drying process is necessary as one-stage drying is insufficient to meet the requirement. A thermal treatment technique is carried out in the pretreatment section when flower waste is exposed to sunlight before it is sent into the biomass conversion process.

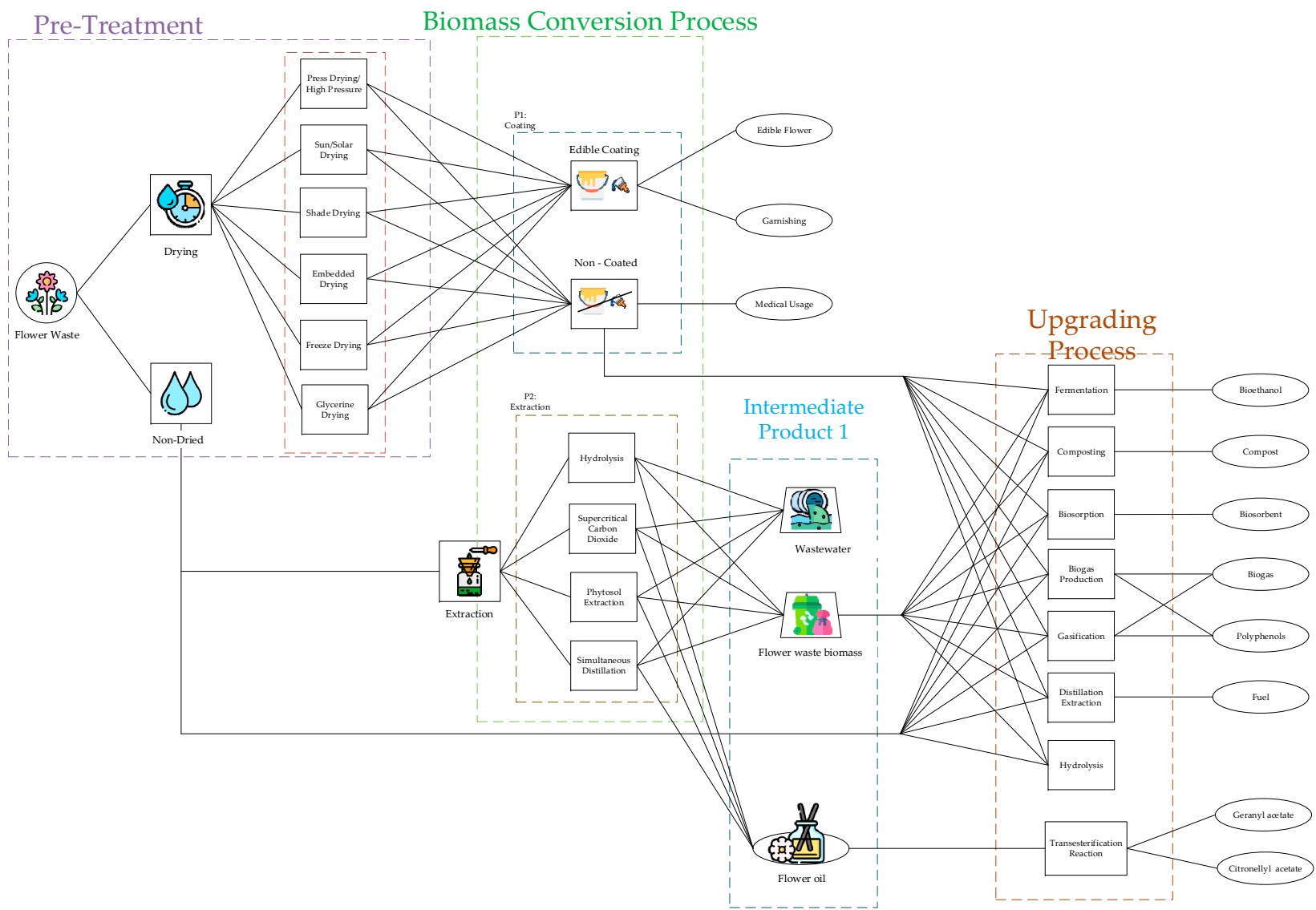

**Figure 2.** Overall superstructure for synthesis of integrated flower.

As shown in Figure 2, flower waste that undergoes pretreatment can then be sent into coating or extraction processes (biomass conversion process). The coating process is to preserve the edible flower with a longer shelf time and to provide mechanical support, preserving the original form of the flower [42]. In this process, retention of quality parameters (i.e., moisture content, colour, flavour, gaseous exchange and oxidation) are considered.

According to Report Linker [44], the U.S. essential oils market by revenue is predicted to grow at a CAGR of 9% by 2026 due to their value for medicinal and recreational purposes. As a result, this superstructure provides another pathway for extraction of flower essential oil. Intermediates like wastewater, biomass waste and flower essential oil are generated after the extraction process. Wastewater will be sent to waste treatment while biomass waste can go through intermediates process to form bioethanol, biogas, geranyl acetate, etc. Based on research work done by Shivaprasad [43], the flower essential oil produced can undergo a transesterification reaction to form geranyl acetate and citronellyl acetate.

According to Statista [1], the flower production reaches 3 million metric tonnes per year, which amounts to 8220 metric tonnes of flowers produced daily. According to Waghmode [45], 40% of India's entire flower production is sold, amounting to 3300 metric tonnes of flowers sold each day. In this case study, it is assumed that half of the flower waste sold is utilised for decorative purposes and for heritage activities; this will result in a total of 1,640,000 kg of waste produced per day, and roughly 68,500 kg every hour. The properties of flower waste, such as density, composition and weight distribution based on flower parts, are summarised in Table 4. Note that three main parts in flower waste are petals, leaves and stems. However, according to case study in India, flower waste left behind from religious offering is frequently incomplete, only consisting of flower petals and leaves. Thus, the stem parts are neglected in the flower-waste weight distribution.

**Table 4.** Properties of flower waste.

| Flower Composition | Petals | Leaves |
|---|---|---|
| Density, kg/m$^3$ [46] | 276 | 276 |
| Weight distribution, % [47] | 38 | 62 |

Performance of the drying processes and the conversion factor required for extraction technologies are summarised in Tables 5 and 6, respectively. Note that for the drying process, the final moisture content is used to compute the mass balance of the integrated flower-waste biorefinery.

**Table 5.** Performance of drying processes.

| Technology/Biorefinery | | Final Moisture Content, % | Intermediates/Product |
|---|---|---|---|
| Drying | Press Drying [36] | 11.55 | Edible flower/garnishing |
| | Sun Drying [36] | 11.55 | |
| | Shade Drying [36] | 11.55 | |
| | Embedded Drying [36] | 11.55 | |
| | Freeze Drying [37] | 24.56 | |
| | Glycerine Drying [37] | 11.55 | |

**Table 6.** Conversion factor of extraction technologies.

| Technology/Biorefinery | | Conversion, % | Intermediates/Product |
|---|---|---|---|
| Extraction | Hydrolysis [43] | 80 | Wastewater, Biomass, and Flower Essential Oil |
| | Supercritical Carbon Dioxid [48] | 95 | |
| | Phytosol Extraction [49] | 87 | |
| | Simultaneous Distillation [50] | 60 | |

**Table 6.** *Cont.*

| Technology/Biorefinery | | Conversion, % | Intermediates/Product |
|---|---|---|---|
| Bioenergy generation | Fermentation [51] | 77 | Bioethanol |
| | Composting [40] | 77 | Compost |
| | Biosorbent [52] | 63 | Biosorbent |
| | Biogas Production and Gasification [41] | 65 | Biogas/Polyphenols |
| | Distillation Extraction [53] | 60 | Fuel |
| | Hydrolysis [43] | 70 | Pectin Oligosaccharide |
| Flower oil further process | Transesterification Reaction [43] | 95 and 70 | Geranyl Acetate and Citronellyl Acetate |

To evaluate the economic performance of the integrated flower-waste biorefinery, the selling price of all final products from different technologies are determined. According to Geitner and Block [54], an average operational hour of a production plant is taken as 8000 h per annum, which is used as the basis of operating hours in this work. To determine the number of batches per year ($n^{dry,annual}$), the total operating hours in a year ($t^{annual}$) can be divided with the time required for one batch of drying process ($t^{dry,batch}$) shown in Equation (10):

$$n^{dry,annual} = \frac{t^{annual}}{t^{dry,batch}} \tag{10}$$

The product cost can then be determined by the ratio of cost per batch with weight per batch for each technology. Table 7 shows the product cost for different drying processes. For processes that are not time-constrained, product cost can be determined by a literature review as listed in Table 8.

**Table 7.** Product cost for drying processes.

| Drying Process | Time Required, h/Batch | Product Cost, USD/kg |
|---|---|---|
| Press drying/High pressure [55] | 2 | 0.50 |
| Sun/Solar drying [36] | 48 | 0.02 |
| Shade drying [36] | 72 | 0.01 |
| Embedded drying [56] | 48 | 0.02 |
| Freeze drying [7] | 240 | 0.004 |
| Glycerine drying [57] | 4 | 0.25 |

**Table 8.** Product cost for upgrading processes.

| Element | Price, USD/kg | Element | Price, USD/kg |
|---|---|---|---|
| Edible flower [58] | 1.45 | Polyphenols [59] | 9.50 |
| Flower for garnishing [60] | 20.00 | Fuel [61] | 2.60 |
| Medical flower [62] | 85.30 | Pectin oligosaccharide [63] | 5.00 |
| Bioethanol [64] | 1.64 | Flower oil [65] | 35.00 |
| Compost [66] | 9.00 | Geranyl acetate [67] | 372.00 |
| Biosorbent [63] | 0.30 | Citronellyl acetate [67] | 311.00 |
| Biogas [63] | 0.33 | | |

To evaluate the environmental impact of the integrated biorefinery, the energy consumption of each technology is required and has been summarised in Table 9.

**Table 9.** Energy required, and carbon emission for different technologies.

| Biorefinery/Technology | | Energy Required, kWh/kg | Carbon Emission, kg $CO_2$/kg Feed |
|---|---|---|---|
| Drying | Press drying [68] | 3.70 | 1.89 |
| | Sun drying | N/A | N/A |
| | Shade drying | N/A | N/A |
| | Embedded drying | N/A | N/A |
| | Freeze drying [69] | 1.60 | 0.82 |
| | Glycerine drying | N/A | N/A |
| Extraction | Hydrolysis [70] | 1.90 | 0.97 |
| | Supercritical $CO_2$ [71] | 0.40 | 0.20 |
| | Phytosol extraction [72] | 0.60 | 0.30 |
| | Simultaneous distillation [73] | 1.50 | 0.002 |
| Bioenergy generation | Fermentation [74] | 0.43 | 0.001 |
| | Composting [75] | 13.14 | 6.70 |
| | Biosorption [75] | 0.43 | 0.22 |
| | Biogas production [76] | 0.32 | 0.0006 |
| | Gasification [77] | 0.30 | 0.15 |
| | Distillation extraction [71] | 1.50 | 0.003 |
| | Hydrolysis [78] | 44.70 | 22.8 |
| Flower oil further process | Transesterification [79] | 6.77 | 3.45 |

Note that most energy required for drying processes are labelled as N/A, as the drying processes do not require electricity nor power consumption. Sun and shade drying processes occur naturally with the aid of heat energy from the sun to reduce the moisture content in flower waste. Meanwhile, for embedded drying and glycerine drying, external material (i.e., silica gel and glycerol solution) is provided to cover the surface of flower waste, and water will be transferred into the silica body and glycerol solution. In this work, it is assumed 0.51 kg $CO_2$ is released when one unit of electricity is utilised [80].

*4.2. Result Analysis*

As mentioned previously, in this work, three scenarios are solved to illustrate the proposed model, which are:

(i)   Maximum economic performance,
(ii)   Minimum environmental impact,
(iii)   Multi-objective optimisation with maximum economic performance and minimum environmental impact.

Scenario 1: Optimisation of flower waste biorefinery with maximum economic performance.

Solving Equation (11) subject to Equations (1)–(8) in a commercial optimisation model, LINGO version 18 with global solver.

$$\text{Maximise } EP \tag{11}$$

Figure 3 illustrates the optimised pathway for scenario 1. Based on the optimisation model, 68,493 kg/h of flower waste undergo a drying process in the pretreatment section to reduce the moisture content to 11.55 wt%. As a result, 23,445 kg/h of flower waste is formed and sent into the biomass conversion process.

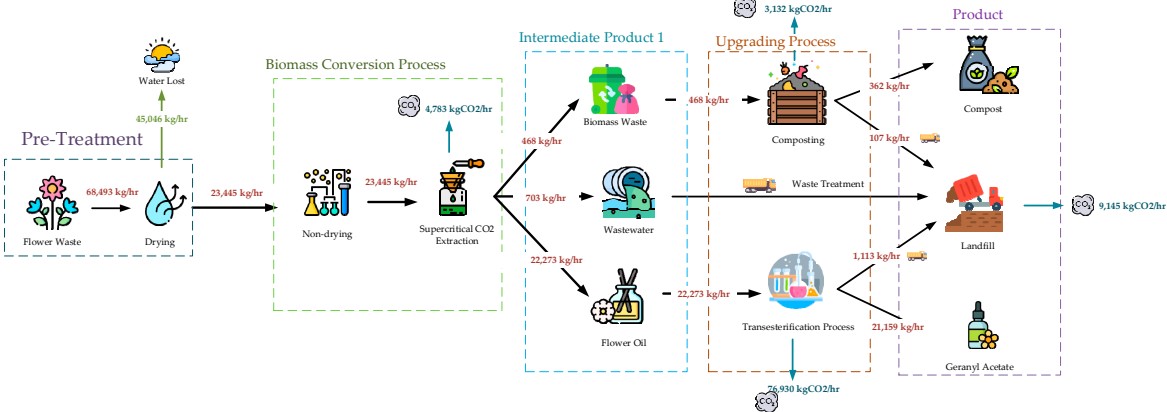

**Figure 3.** Optimisation based on maximum economic potential—scenario 1.

Next, flower waste is extracted with a supercritical carbon dioxide extraction method to be further converted into different intermediate products such as biomass waste, wastewater and flower essential oil. The amount of intermediate product is 468 kg/h, 703 kg/h and 22,273 kg/h, respectively.

Then, both biomass waste and wastewater are sent to the upgrading process and waste treatment process. The composting method is selected among the upgrading processes (e.g., fermentation, biogas production, biosorbent production, etc.) to form 362 kg/h of compost. Flower essential oil is converted into 21,159 kg/h of geranyl acetate via the transesterification process. The rest of the waste produced is sent to a landfill. The overall economic potential and total carbon emission in scenario 1 are summarised in Table 10.

**Table 10.** Economic potential and environmental impact in scenario 1.

| Total Revenue | Total Carbon Emission |
|---|---|
| USD 7,874,455 | 93,999 $kgCO_2/h$ |

Scenario 1 shows that the pathways chosen to have significantly high revenue, but the total carbon dioxide released is extremely high as the environmental impact is being neglected. The carbon generated in scenario 1 is mostly from the transesterification process, which is 76,930 kg $CO_2/h$, as this process requires the use of machinery and electricity.

Table 11 shows the parameters that affect revenue in scenario 1. The waste generated in scenario 1 is relatively less, around 1924 kg/h. This is due to the biomass formed after extraction process being sent to the upgrading process to form other value-added products, which are compost and geranyl acetate, both with extremely high selling costs. Thus, the main revenue in scenario 1 only depends on the product selling price; treatment cost in this scenario does not have a big impact on the economic potential.

**Table 11.** Parameters that affect revenue.

| Product Selling Price | Treatment Cost |
|---|---|
| USD 7,874,474 | 1924 $ |

Scenario 2: Optimisation of flower waste biorefinery with minimum environmental impact.

The synthesis of an integrated flower-waste biorefinery in scenario 2 is solved based on the optimisation objective listed in Equation (12). With this, the pathway of choosing the minimum environmental is determined.

$$\text{Miminise } TCE \tag{12}$$

Figure 4 shows the summarised refinery pathway based on the lowest environmental impact generated. First, 68,493 kg/h of flower waste will undergo a two-stage drying; first, sun drying to produce 23,445 kg/h of flower waste and, second, glycerine drying to form 15,904 kg/h of flower waste. Next, in the biomass conversion process, the flowers are coated to form the exact amount of flowers for decoration purposes. The overall economic potential and total carbon emission in scenario 2 is summarised in Table 12.

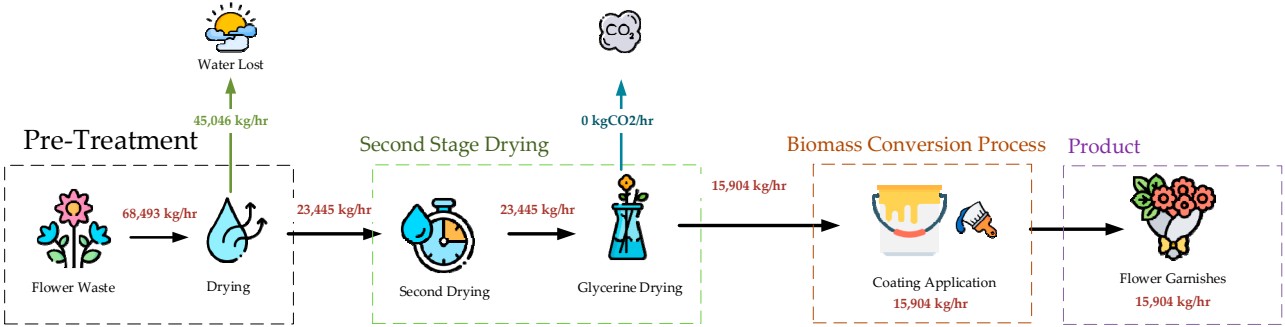

**Figure 4.** Optimisation based on minimum environmental impact—scenario 2.

**Table 12.** Economic potential and environmental impact in scenario 2.

| Total Revenue | Total Carbon Emission |
|---|---|
| USD 11,927 | 0.01 kg $CO_2$/h |

According to optimised results, the flower-waste mass flow is accumulated and sent to the glycerine drying and coating process, which requires no energy consumption nor release of carbon dioxide into the atmosphere. In this case, zero emissions can be demonstrated. Table 12 shows that the economic potential and environmental impact of optimised pathway in scenario 2. Table 13 shows carbon emission from landfill.

**Table 13.** Carbon emission from landfill.

| Carbon Emission, kg $CO_2$/h |
|---|
| Landfill |
| 0.01 |

Scenario 3: Optimisation of flower waste biorefinery with maximum economic potential and minimum environmental impact.

Scenario 1 and scenario 2 can only achieve one objective at a time. Thus, fuzzy optimisation is adopted for the synthesis of an integrated flower-waste biorefinery that considers both objectives simultaneously. The value of total revenue and total carbon emission from both scenario 1 and scenario 2 shown in Table 14 are taken as the upper and lower limits for fuzzy optimization equations shown in Equations (13) and (14), where $X^U$, $X^L$, $Y^U$ and $Y^L$ are the upper and lower limits of total revenue and total carbon emissions, respectively.

**Table 14.** Parameters required for fuzzy optimisation.

| | Total Revenue | Parameters | Total Carbon Emission | Parameters |
|---|---|---|---|---|
| **Scenario 1** | USD 7,874,474 | $X^U$ | 93,999 kg $CO_2$/h | $Y^U$ |
| **Scenario 2** | USD 11,927 | $X^L$ | 0.01 kg $CO_2$/h | $Y^L$ |

Equations (13) and (14) describe the basis of fuzzy:

$$\frac{TR - X^L}{X^U - X^L} \geq \lambda \tag{13}$$

$$\frac{Y^U - TCE}{Y^U - Y^L} \geq \lambda \tag{14}$$

with this, the pathway of choosing the minimum environmental is determined as:

$$\text{Minimise } \lambda \tag{15}$$

Figure 5 shows the optimised pathway for scenario 3. The pathways chosen by Lingo are similar to the combination of both scenarios 1 and 2, but with a different mass distribution and upgrading process. The amount of pre-dried flower waste in pretreatment is first divided into half and sent to two pathways. First, 11,521 kg/h of pre-dried flower undergo another stage of drying and coating process to form 7815 kg/h of flowers for decorative purposes. Second, another 11,923 kg/h of pre-dried flower is sent to a supercritical carbon dioxide extraction process to form 238 kg/h of biomass waste, 358 kg/h of wastewater and 11,327 kg/h of flower essential oil as intermediate products. Next, biomass waste undergoes fermentation to form 184 kg/h of bioethanol; flower essential oil goes through a transesterification process and eventually forms 10,761 kg/h of geranyl acetate.

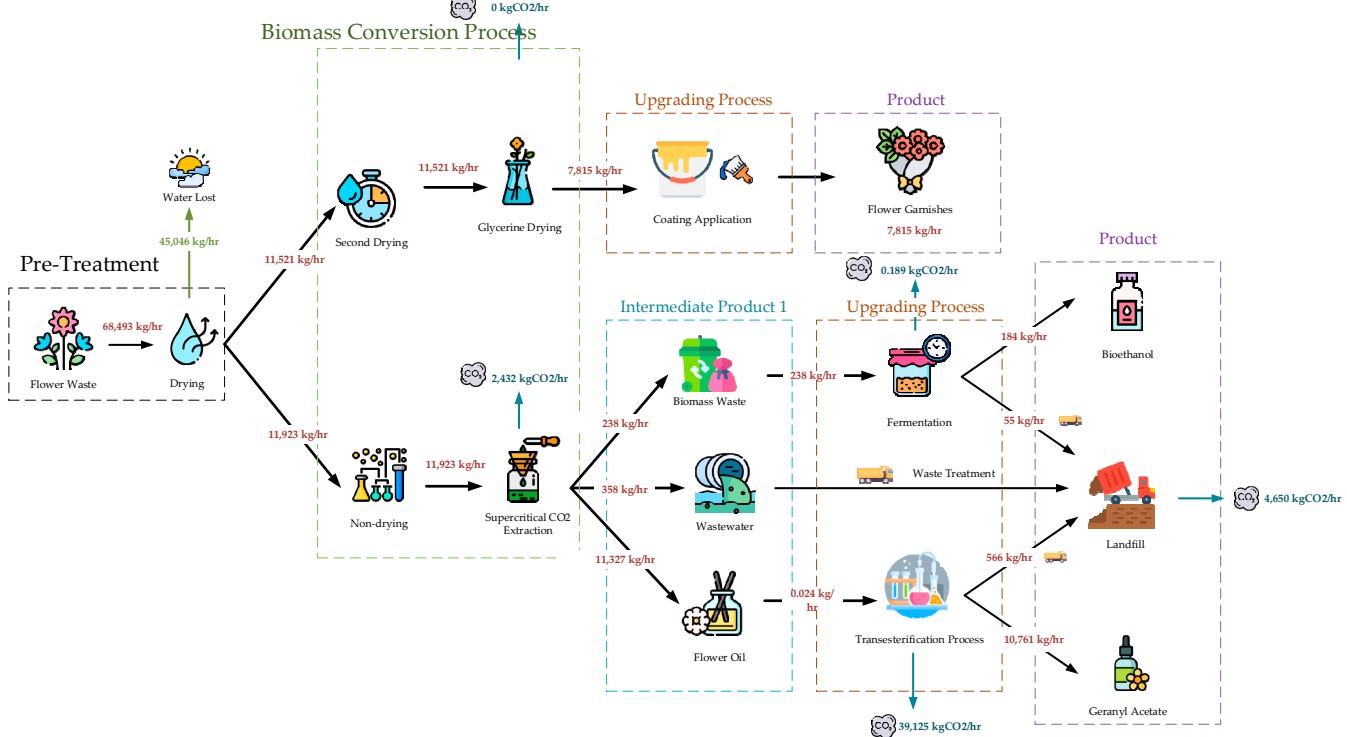

**Figure 5.** Optimisation based on maximum economic potential and minimum environmental impact—scenario 3.

The overall economic potential and total carbon emission in scenario 3 is summarised in Table 15. This tables shows that both revenue and environmental impact have achieved a balance between scenarios 1 and 2.

**Table 15.** Economic potential and environment impact in scenario 3.

| Total Revenue | Total Environmental Impact |
|:---:|:---:|
| USD 4,009,320 | 46,209 kg $CO_2$/h |

## 5. Conclusions

This paper presented a generic mathematical optimisation approach for synthesis of a sustainable integrated flower-waste biorefinery. An Indian case study was solved to illustrate the proposed optimisation model. Multi-objective optimisation (fuzzy optimisation) is adopted as a trade-off between the economic and environmental performances of the integrated biorefinery. Note that the proposed approach can be modified to develop an integrated flower-waste biorefinery based on different geographical constraints and availability of flower waste. This proposed approach demonstrates a preliminary feasibility study of integrated biorefineries based on different parameters, such as conversion ratio, electricity demand product unit cost, etc. Future work involves integrating sensitivity analyses to the above approach on selected parameters to understand their economic impact towards the selection of technologies and processes.

**Author Contributions:** Conceptualization, E.H.Y.C., V.A., L.Y.N., P.S. and D.K.S.N.; methodology, E.H.Y.C., V.A., L.Y.N. and D.K.S.N.; software, E.H.Y.C. and D.K.S.N.; validation, V.A., L.Y.N., P.S. and D.K.S.N.; formal analysis, E.H.Y.C.; investigation, E.H.Y.C.; resources, D.K.S.N.; data curation, E.H.Y.C.; writing—original draft preparation, E.H.Y.C.; writing—review and editing, E.H.Y.C., V.A., L.Y.N., P.S. and D.K.S.N.; visualization, E.H.Y.C.; supervision, V.A., L.Y.N., P.S. and D.K.S.N.; project administration, V.A., L.Y.N. and D.K.S.N.; funding acquisition, D.K.S.N. All authors have read and agreed to the published version of the manuscript.

**Funding:** This research received no external funding.

**Acknowledgments:** This research was funded by Royal Academy of Engineering under the Frontiers of Development scheme.

**Conflicts of Interest:** The authors declare no conflict of interest.

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
