# Peer review of "Synthesis of Integrated Flower Waste Biorefinery: Multi-Objective Optimisation with Economic and Environmental Consideration"

_processes, doi:10.3390/pr10112240_

Round 1

Reviewer 1 Report

The logic of the article needs to be improved, in the introduction. In the introduction, the author does not well express the innovation and purpose of the research. How does this study stand out compared to similar studies? These need to be more highlighted. Some tables are non-standard. The author needs to improve it.Introduction does not highlight literature gap properly in my opinion. There are several significant findings not approached in the state-of-the-art description in this manuscript . You could analyze more thoroughly also the pre-treatment methods of floral waste. Please add also the releant to pre-treatment metods summarizing work of DOI: 10.3390/su132212810.

The topic of the manuscript is interesting,  and well-timed though the title does not clearly present the topic of this work . In the beginning, you do not elucidate what kind of paper it is (article, short note, case study ) to be clear to the readers. 

The language in this manuscript needs to be revised. There were many grammar mistakes, such as the wrong tense and a set phrase. The article needs to be carefully polished in spoken English.The conclusion is too short and the author should  improve the logic of the conclusion. The discussion of the results is missing.

The details of how the various biomass components and type of flowers waste for instance jasmine, rossele etc effect the the quality of gas produced and how thiw effect the economical analysis model.  It is not only lignin and extractives, but there are also other biomass components which impact the quality of gas.

It is fact that for  the extraction of dyes, flowers were dried and grinded and were dissolved in solvents such as ethanol, methanol and hexane.The use of these chemical solvents have environmental impact and the cost for managment and recycle of these solvents is high or not?

It is also not clear how the flower waste biomas, the part of flower and the physico-chemical parameters of floral waste like temperature,pH, electrical conductivity moisture content  impacts the energy consumption of the process. 

Author Response

Thanks for the review comments. Please find the response to the reviewer as attached. 

Reviewer 2 Report

This study presents the synthesis of an integrated flower waste biorefinery having economic and environmental considerations. Three scenarios were analysed in the case study: (i) maximum economic performance (ii) minimum environmental impact and (iii) multi-objective optimisation with maximum economic performance and minimum environmental impact. This is a very good and interesting study. The Introductory section is well written. The results are clear and were well represented on the flow diagrams and tables. The study was compared with past studies. The effort of the authors is commendable but minor improvements are needed.

1.      Check for typographical errors throughout the manuscript. For example, Line 38. “source od waste”, Line 42 and Line 219, correct them. Sometimes, you write synthesizing or synthesizing.

2.      The abstract should be in this order: background, aim, methods, results and conclusion.

3.      Some points on the conclusion are also important for the abstract.

Author Response

Thank you very much for the comments. Please find the response to reviewer as attached. 

Round 2

Reviewer 1 Report

 Τhe new version of the manuscript after the first revision is significantly improved. Many of the comments and remarks were successfully considered, and included in the rewritting and reorganizing of the manuscript.The introduction and literature review has been improved.The impact of flower waste towards the energy consumption has been enhanced.